# Mechanical High-Temperature Properties and Damage Behavior of Coarse-Grained Alumina Refractory Metal Composites

**DOI:** 10.3390/ma12233927

**Published:** 2019-11-27

**Authors:** Anja Weidner, Yvonne Ranglack-Klemm, Tilo Zienert, Christos G. Aneziris, Horst Biermann

**Affiliations:** 1Institute of Materials Engineering, Technische Universität Bergakademie Freiberg, 09599 Freiberg, Germany; klemm_yvonne@web.de (Y.R.-K.); biermann@ww.tu-freiberg.de (H.B.); 2Institute of Ceramic, Glass and Construction Materials, Technische Universität Bergakademie Freiberg, 09599 Freiberg, Germany; tilo.zienert@ikgb.tu-freiberg.de (T.Z.); aneziris@ikgb.tu-freiberg.de (C.G.A.)

**Keywords:** refractory composites, high-temperature, mechanical properties, microstructure

## Abstract

The present study provides the mechanical properties of a new generation of refractory composites based on coarse-grained Al_2_O_3_ ceramic and the refractory metals Nb and Ta. The materials were manufactured by refractory castable technology and subsequently sintered at 1600 °C for 4 h. The mechanical properties and the damage behavior of the coarse-grained refractory composites were investigated at high temperatures between 1300 and 1500 °C. The compressive strength is given as a function of temperature for materials with two different volume fractions of the refractory metals Ta and Nb. It is demonstrated that these refractory composites do not fail in a completely brittle manner in the studied temperature range. The compressive strength for all materials significantly decreases with increasing temperature. Failure occurred due to the formation of cracks along the ceramic/metal interfaces of the coarse-grained Al_2_O_3_ particles. In microstructural observations of sintered specimens, the formation of tantalates, as well as niobium oxides, were observed. The lower compressive strength of coarse-grained Nb-Al_2_O_3_ refractory composites compared to Ta-Al_2_O_3_ is probably attributed to the formation of niobium oxides. The formation of tantalates, however, seems to have no detrimental effect on compressive strength.

## 1. Introduction

Ceramic–metal composites benefit from the combination of a high melting point, hardness and the chemical stability of ceramics, with the high toughness and ductility of metals. In general, the upper limit of the application temperature of such composites is restricted by the melting point of the metal, the reaction between the metal and ceramic particles, the chemical interaction with the environment, or the thermal mismatch between the ceramic and the metallic phases. Therefore, the use of refractory metals with a high melting point could make such composites applicable at even higher temperatures [1].

Typical refractory metals with a melting temperature above that of alumina (T_m_ = 2054 °C), are Mo, Nb, W, Ta, Hf, Tc, Re, Os, and Ir. Among them, Mo, Nb, W, and Ta are the most abundant [2,3]. Niobium and tantalum exhibit a similar coefficient of thermal expansion (CTE) as alumina in a wide temperature range [4]. For instance, Kirby et al. [4] showed that, between 1000 and 2000 °C, a linear increase of CTE occurred for all three materials, with 25–34 × 10^−6^·K^−1^ for alumina, 24–30 × 10^−6^·K^−1^ for niobium, and 20–25 × 10^−6^·K^−1^ for tantalum. Accordingly, composites of alumina with tantalum or niobium could be promising candidates for thermal shock resistant materials, as shown by some research groups [5,6,7,8,9,10,11]. Moreover, the interface between Nb and Al_2_O_3_ should be stable at high temperatures in an inert atmosphere [12,13,14,15,16,17,18,19,20]. Mechanical properties [6,7,10,11], wear behavior [21], chemical stability [19,22] and thermal conductivity [23] of the sintered mixtures of Nb and Al_2_O_3_ were reported only for components made of very fine powders. In particular, fine-grained tantalum-alumina composites were studied as a potential material for femoral head in total hip arthroplasty [24].

In the past, several investigations concerning high-temperature properties (creep, fracture toughness) were performed for Al_2_O_3_ ceramics [25,26], as well as refractory ceramics with up to 50% niobium [8,9,19,27,28,29]. In these papers, different damage mechanisms were discussed, e.g., ceramic-metal interface decohesion [8,29], fracture of ceramic particles [29], and retardation of crack propagation by a high-volume fraction of niobium [8,29], as well as the resulting high fracture toughness [9]. Furthermore, the formation of different intermetallic phases (AlNb_2_, Al_3_Nb) and niobium oxide (NbO) during (pressureless) sintering was reported [19,29]. However, the materials described in these papers are, without exception, fine-grained materials (particle size: Al_2_O_3_ < 100 µm, Nb < 50 µm) exhibiting high densities (>95%) [8] after the sintering or hot-pressing process.

In addition, there are studies on yttrium-stabilized, tetragonal zirconium dioxide, with 20 vol. % Nb [30,31], and zirconium dioxide/alumina, with 20 vol. % Ta [32], as well as zirconium dioxide, with 20 vol. % Nb and 20 vol. % Ta [33]. These materials were also very fine-grained composites with micro- to nano-sized particles. For this group of materials, improved fracture toughness was observed as reported for Ta- or Nb-added ceramics. The toughness results from two factors: (i) energy dissipation by the plastic deformation of the metallic components (Nb, Ta), and (ii) tetragonal to monoclinic phase transformation of the zirconium dioxide, which is accompanied by a volume expansion (approx. 4%) and generates compressive stresses.

Investigations on damage and crack growth at room temperature were carried out in several studies [8,28,32,33], on the basis of cracks generated by Vickers’ indentation on the tensile-loaded surface of bending specimens. The crack-bridging effect, caused by plastic deformation of the metallic phases, was shown both for Ta and Nb.

Coarse-grained ceramic refractories are used in favor of fine-grained ceramic materials in high-temperature and metallurgical applications, due to their relatively high resistance to creep, thermal shock and corrosion resistance [34,35,36,37,38]. Furthermore, it was shown by Zienert et al. [39] that the shrinkage can be even further reduced by adding high melting refractory metals as fine matrix component. The sum of these promising results regarding the shrinkage behavior in combination with both good resistance to thermal shock, as well as the excellent electrical and thermal properties of these materials, offer the opportunity to selectively design individual parts of large-sized construction components. Thus, large components can be manufactured with negligible thermal stresses due to the negligible shrinkage behavior during sintering.

The aim of the present paper is to investigate a new generation of coarse-grained refractory composites [39] based on pre-synthesized coarse grains of a ceramic (Al_2_O_3_) and refractory metals (Nb, Ta). Considering their potential applications as, e.g., electrically controlled heat shields or non-wetting, electrically heatable crucible materials, experimental studies on their mechanical properties at high temperatures (in particular, compressive strength, creep, and stress-relaxation, as well as damage and fracture behavior) are necessary. Therefore, high-temperature mechanical properties under compressive loading of the coarse-grained refractory composites, based on coarse-grained alumina (up to 5 mm) and fine-grained niobium and tantalum, were investigated. The refractory composites were manufactured by powder metallurgical route with different volume fractions of tantalum or niobium. The mechanical properties were tested at 1300, 1400 and 1500 °C. In addition, compressive stress relaxation tests were performed. The microstructure of the tested materials was examined by scanning electron microscopy (SEM) and energy-dispersive spectroscopy (EDS).

## 2. Materials and Methods

### 2.1. Materials and Preparation

The materials under investigation were coarse-grained refractory composites based on coarse-grained alumina and fine-grained refractory metals niobium and tantalum. The following raw materials were used: tabular alumina T60/64 (Almatis GmbH, Frankfurt, Germany), reactive alumina CL370 (Almatis GmbH), alumina Martoxid MR-70 (Martinswerk GmbH, Bergheim, Germany), re-hydratable alumina binder Alphabond 300 (Almatis GmbH), dispersing aluminas ADS-1 and ADW-1 (Almatis GmbH), Nb powder (EWG E. Wagener GmbH, Heimsheim, Germany), Ta powder (haines & maassen Metallhandelsgesellschaft mbH, Bonn, Germany). The refractory metals were used as powders with a particle classification d ≤ 75 µm, whereas the alumina used had different particle classifications (0–20 µm: 10 vol. %, 0–45 µm: 11 vol. %, 0–0.5 mm: 21 vol. %, 0.5–1 mm: 10 vol. %, 1–3 mm: 13 vol. %, and 2–5 mm: 24 vol. %). The two fractions with the smallest alumina particles (0–20 µm: 10 vol. %, 0–45 µm: 11 vol. %) were replaced by fine powder fractions (0–20 µm) of Ta and Nb, respectively, both with a purity of 99.95%. If both fractions of fine alumina particles were replaced by refractory metals composites with 21 vol. % of refractory metal resulted, whereas composites with 11 vol. % were obtained by replacing only one fraction of alumina (0–45 µm). Thus, Ta-Al_2_O_3_ composites and Nb-Al_2_O_3_ composites were manufactured with both 11 vol. % and 21 vol. % of refractory metals. The coarse-grained refractory composites were fabricated using a refractory castable technology consisting of a mixing procedure of powder mixtures with a volume of ≈350 cm^3^ in a concrete mixer (ToniMAX, Toni Baustoffprüfsysteme GmbH, Berlin, Germany) using 5 mass% water and 1 mass% of dispersing alumina as additives. Produced castable blends were cast into prismatic (150 × 25 × 25 mm³) sample molds and remained there for 1 to 3 days. Hereafter, the samples were taken out and dried in air at 120 °C for 24 h, followed by a pressureless sintering regime under argon atmosphere at 1600 °C for 4 h. Heating rates of 3 K/min up to a temperature of 500 °C, and 10 K/min up to the sintering temperature of 1600 °C, were applied. Details of the chemical composition of the raw materials, the castable recipe and the castable technology were reported recently in [39]. Cylinders with a diameter of 15 mm and a height of 25 mm were prepared for the high-temperature mechanical tests from the sintered prism by hollow drilling. Since the mechanical tests (compression, stress–relaxation) require plane–parallel front faces, these faces were mechanically grinded using a special tool designed for this purpose (see Figure 1a). Therefore, the specimens were fixed by a semi-circular clamping part with well-defined geometry according to the diameter of the specimens. Subsequently, the upper and lower front faces (after turning the tool by 180° around the axis perpendicular to the cylinder axis) were ground. The porosity was determined before and after mechanical tests using both mercury porosimetry and the Archimedes’ method.

### 2.2. Mechanical Testing

To investigate the mechanical behavior of the coarse-grained refractory composites, quasi-static compression and stress–relaxation tests were carried out on an electro-mechanical high-temperature 20 kN testing machine (Zwick/Roell, Neu-Ulm, Germany), as shown in Figure 1b. The compression tests were performed with a crosshead displacement rate of 15 µm/s which results in an initial strain rate of about 7.5 × 10^−4^·s^−1^. The tests were carried out under argon atmosphere (gas-chamber: Maytec, Singen, Germany) after two evacuation cycles, in order to avoid oxidation of the metallic phases of the specimens. A detailed view of the specimen set-up and the heating system is shown in Figure 1c. The cylindrical specimen (1) is placed in a susceptor cage (6) between two Mo-based susceptors (5) (TZM alloy) to set a homogenous temperature distribution. The heating was performed by a water-cooled copper induction coil (2) controlled by a middle-frequency generator (Hüttinger HF 5010, Freiburg, Germany) with heating rates up to 20 K/s. Strain was measured with a high-temperature extensometer system with alumina rods (3). Si_3_N_4_-pistons (4) transfer the load from the pressure tubes to the shown setup. The specimens were tested at 1300, 1400, and 1500 °C. For the metallic components, this corresponds to a homologous temperature of 0.43 ≤ T_hom_ ≤ 0.49 for Ta, and 0.52 ≤ T_hom_ ≤ 0.60 for Nb. The temperature was measured on the surface of the specimens using a pyrometer Metis MS09 (Sensortherm, Sulzbach, Germany). A dwell time of 5 min was allowed prior to the tests in order to guarantee a homogenous temperature distribution and to minimize the effects of thermal expansion of the susceptors on the strain measurement. One specimen was tested for each material composition (11 vol. % Ta or Nb and 21 vol. % Ta or Nb, respectively) and each temperature. It should be noted here that the authors are aware of scatter in mechanical data of ceramic materials. However, it is known from other ceramic materials, such as carbon-bonded magnesia (MgO-C), that the scatter of data decreases with an increase in the “pseudo ductility” of these materials at high temperatures, depending on the content of graphite. However, for these new coarse-grained Ta-Al_2_O_3_ and Nb-Al_2_O_3,_ the Weibull distribution of the mechanical data cannot be provided due to limited material.

Stress relaxation tests were performed under compressive load at the same temperatures. The compressive load for the relaxation tests was chosen according to 80% of the maximum load achieved during the high-temperature compression tests.

### 2.3. Microstructural Investigations

The microstructure of the refractory composites was investigated before and after the compression tests, using a Scanning Electron Microscope (SEM), (Mira 3 XMU, Tescan, Brno, Czech Republic). SEM micrographs were taken in Secondary Electron (SE) or Back-Scattered Electron (BSE) contrast. In addition, Energy-Dispersive Spectroscopy (EDS) was performed in terms of point and line analysis, in order to study possible reactions of the refractory metals with alumina using EDS detector (Apollo 10 mm²) and acquisition software from EDAX.

## 3. Results and Discussion

### 3.1. Initial Microstructure of Refractory Metal-Alumina Composites

The microstructure of the specimens after the sinter process was investigated on cylindrical specimens cut parallel to the cylinder axis. Figure 2 shows exemplarily SEM micrographs of specimens, with 21 vol. % Ta and Nb, respectively. In Figure 2a, the coarse-grained fraction of alumina particles (>1 mm) is clearly visible, with pores remaining from the manufacturing process. Both the Ta and the Nb particles exhibit a good bonding with the alumina particles. However, in both materials, a reaction between alumina and refractory metals seems to occur during the sintering process. Clear indicators for this assumption are the different grey levels found both beside Ta particles (Figure 2b), as well as Nb particles (Figure 2c).

To study the reaction of Ta and Nb with Al_2_O_3_, EDS line measurements were performed on both microstructures. The results are summarized in Figure 3 and Figure 4, for Nb and Ta refractory composites, respectively. It is clearly visible that, in the dark-grey area of Nb particles (cf. Figure 3a), an enrichment of oxygen was measured by EDS, whereas, in the brighter-grey area, nearly 100% of Nb was measured (cf. Figure 3b). This enrichment of oxygen in Nb was observed frequently in the microstructure and is most probably the result of the sintering process.

Similar behavior was found for Ta particles. Here, also different grey levels in the SEM micrograph (see Figure 4a) serve as an indicator for the formation of a new phase beside pure Ta and alumina. In contrast to Nb refractory composites, here, an enrichment not only of oxygen but also of aluminum was detected, as the EDS line profile shows in Figure 4b. Thus, a sequence from pure tantalum (bright-grey; 1) over pure alumina (dark-grey) to an area composed of O, Ta and Al (light-grey) to pure Ta again (bright-grey), ending in pure alumina (2), was detected. This frequently observed enrichment of oxygen and aluminum in Ta-particles is mostly attributed, as mentioned above, to the consequence of reactions during the sintering process.

It is assumed that this phase could be representative of the group of aluminum tantalates [40,41]. Tantalum has a high affinity for oxygen [42,43]. Tantalum powders, therefore, form a tantalum-(V)-oxide (Ta_2_O_5_) [42,43] at the surface. At higher temperatures (approx. 1300 °C), this oxide can react with Al_2_O_3_ to aluminum tantalate AlTaO_4_ [40,41,44] (light-grey in Figure 4a). However, the ternary system Ta-Al-O is not yet well investigated regarding both the occurring phase equilibria as well as the thermodynamics. King et al. [45] and Roth and Waring [46] showed, for the subsystem Ta_2_O_5_-Al_2_O_3,_ a solubility of about 10 mol. % Al_2_O_3_ in Ta_2_O_5_, and the formation of an intermediate phase –TaAlO_4_—with an extended range of solid solution, as reported by Yamaguchi et al. [47]. In addition, Huang et al. [24] showed for hot-pressed specimens (1450 °C, 1650 °C) that, for these temperatures, a diffusion of oxygen in Ta as well as, at 1650 °C, a diffusion of Al in tantalum, occurred, since the formation of an interphase layer was observed. However, a diffusion of Ta in Al_2_O_3_ was not observed [24]. In similar way, the formation of aluminum niobates of the form AlNbO_4_ was expected [44] for the system Al_2_O_3_-Nb. However, here the formation of niobium–oxide was observed, as reported earlier by other groups [19,29]. A maximum solubility of oxygen in niobium of 0.9 at. %, and the formation of different types of stable niobium oxides (NbO, NbO_2_, Nb_2_O_5_ and Nb_12_O_29_ [48]) were reported. The formation of different intermetallic phases like niobium–aluminides of types AlNb_2_ and Al_3_Nb, as mentioned in [19,29], was not observed in our SEM investigations.

### 3.2. High-Temperature Compression Behavior

Figure 5 shows the results of the high-temperature compression tests on the composites. Figure 5a,b show the stress vs. strain curves for 11 vol. % (a) and 21 vol. % (b) niobium, and Figure 5c,d for tantalum. It is noteworthy that the materials show at least some limited ductility and do not fail in a brittle manner after maximum strength, despite the high-volume fraction of coarse-grained alumina. In all cases, the stress–strain curves exhibit the highest strength and elongation for 1300 °C. The maximum strength decreases significantly with an increase in temperature. Furthermore, the increase in volume fraction of the refractory metals results in a significant decrease in strength. However, the ductility seems to be comparable. Figure 6 shows the dependence of maximum compression strength on temperature for all tested materials. In general, the compressive strength of the materials with 11 vol. % of Ta or Nb is higher compared to the materials with higher volume fraction of metals. Furthermore, the compressive strength decreases for all materials with an increase in temperature. For the lower volume fraction, the Ta-composite exhibits a higher strength than that with Nb. For the higher volume fraction, both composite materials show comparable strength values. However, the decrease in compressive strength with an increase in temperature is less pronounced for materials with a higher volume fraction of refractory metals. The elongation to failure decreases with an increase in temperature. Furthermore, it should be noted that the porosity of all materials is comparable before and after mechanical testing, and lies in the order of about 17%, which means that no consolidation occurred during the high-temperature compression tests.

The macroscopic damage patterns of coarse-grained Ta-Al_2_O_3_ and Nb-Al_2_O_3_ specimens deformed at 1300 °C are summarized in Figure 7. The upper row of Figure 7 shows one of the two front faces, while the lower row shows side views of the cylindrical specimens. The coarse-grained Al_2_O_3_ with grain sizes up to 5 mm is clearly visible on the end faces. In addition, macro-porosity is visible. From the side views of the specimens it can be seen that the cracks run predominantly along the ceramic/metal interfaces. Furthermore, the specimens did not fail completely in a brittle manner; instead, some remaining ductility occurred. However, in all cases mechanical spallation of outer areas was observed.

The damage behavior was also investigated by SEM after the compression tests. Figure 8a,b show specimens with 11 vol. % Nb (a) and Ta (b), respectively, after compression tests at 1500 °C. The pronounced porosity and the severe damage of the microstructure with cracks running from pores along the interfaces between the large Al_2_O_3_ grains and the grains of the refractory metal can be clearly seen in both specimens. In addition, Figure 8d–f also reveal the damage behavior of the Nb particles. In particular, Nb particles with enrichment of oxygen seem to be favored for crack initiation. Thus, many tiny cracks (marked by black arrows) were observed in these areas indicating an embrittlement of the Nb particles by the formation of niobium oxides, which can be the reason for the lower strength of coarse-grained Nb-Al_2_O_3_ refractory composites compared to Ta-Al_2_O_3_ composites. The formation of fine Ta fringes along the grain boundaries of Al_2_O_3_ was observed (Figure 8c), indicating the diffusion of the refractory metals along the grain boundaries during the sintering process.

Thus, the chemical bonding between alumina and refractory metals contributes to good mechanical properties in the Ta-Al_2_O_3_ and Nb-Al_2_O_3_ refractory composites compared to carbon-bonded oxides (MgO-C or Al_2_O_3_-C), where only a mechanical clamping between square-edged oxide particles contributes to the strength of these materials, since no chemical reaction between oxides and graphite occurs [49].

### 3.3. Stress-Relaxation Tests

In addition to the high-temperature compression tests, stress relaxation experiments were carried out at 1300 °C and 1500 °C. The initial compressive stresses were set to 80% of the maximum compressive stresses determined from the respective quasi-static compression tests (cf. Figure 5). The results of the relaxation tests are summarized in Figure 9. The samples with 11 vol. % Ta show the highest resistance to stress relaxation at 1300 °C, which corresponds to a homologous temperature of the metallic phase of T_hom_ = 0.43. Within the first 120 s, a stress drop of approx. 5–6 MPa occurs (approx. 10% of the stress). After that, only a small decrease in stress is observed, due to creep processes. For the higher testing temperature (i.e., T_hom_ = 0.49) a much stronger relaxation is observed (approx. 10 MPa stress drop, i.e., up to 30%). For the composite materials with 11 vol. % and 21 vol. % Nb, respectively, at 1300 °C and 1500 °C (equivalent to the homologous temperatures of Nb of T_hom_ = 0.52 and 0.6), a more pronounced relaxation behavior was observed within the first 3–4 min.

Specimens with 21 vol. % Nb show a stress relaxation only within the first few seconds, and a horizontal curve develops in the further course of the test, i.e., there are almost no further relaxation phenomena. The reason for the significant stress relaxation in all specimens within the first seconds or minutes is probably due to plastic deformation/dislocation rearrangement within the metallic components of the composite materials. It is highly probable that there are changes in the dislocation arrangement introduced by the initial compressive loading.

## 4. Summary

High-temperature mechanical properties were determined for the new class of coarse-grained refractory composites, based on Al_2_O_3_ and 11 or 21 vol. % of refractory metals Ta and Nb, respectively. It was demonstrated that these coarse-grained refractory composites do not fail in a completely brittle manner during compression tests between 1300 and 1500 °C. The compressive strength for all materials significantly decreases with an increase in temperature. Severe damage due to cracks running along the ceramic/metal interfaces of the coarse-grained Al_2_O_3_ particles was observed. The lower compressive strength of coarse-grained Nb-Al_2_O_3_ refractory composites compared to Ta-Al_2_O_3_ ones is probably attributed to the formation of niobium–oxides during the 4 h sinter process at 1600 °C. In contrast, the formation of tantalates seems not to have a detrimental effect on the compressive strength.

## Figures and Tables

**Figure 1 materials-12-03927-f001:**
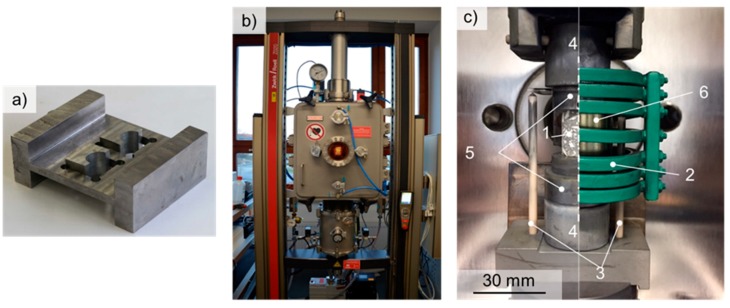
Setup for high-temperature testing of refractory metal-alumina composites. (**a**) Tool for manufacturing of plane–parallel front faces of cylindrical specimens for compression tests. (**b**) Testing machine with protective gas chamber. (**c**) Setup for compression tests. (1) Specimen. (2) Induction coil. (3) Alumina extensometer rods. (4) Si_3_N_4_ punches. (5) Susceptors. (6) Susceptor cage.

**Figure 2 materials-12-03927-f002:**
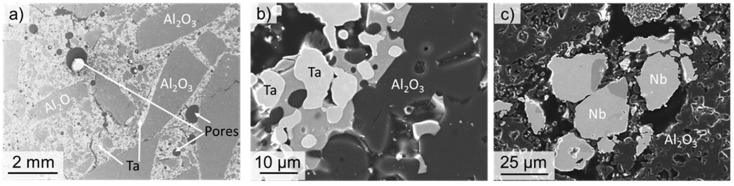
Microstructure of coarse-grained refractory metal-alumina composites after sintering at 1600 °C for 4 h. (**a**) Overview on microstructure of a specimen with 21 vol. % of Ta. (**b,c**) Details of microstructures of specimens with 21 vol. % Ta (**b**) and Nb (**c**), respectively, showing good bonding between alumina particles and refractory metal Ta (**b**) and Nb (**c**). In both cases, additional phases seem to develop during sintering. All micrographs in SE contrast.

**Figure 3 materials-12-03927-f003:**
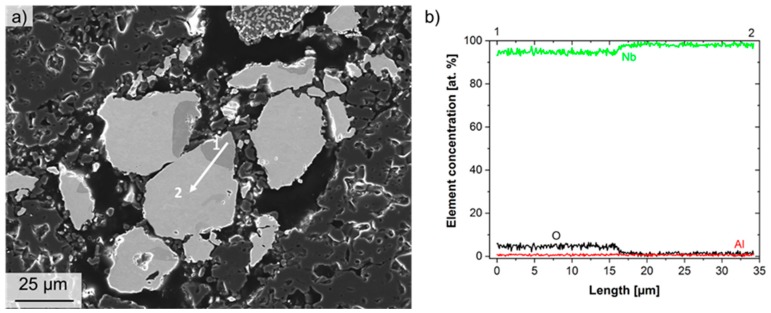
Results of energy-dispersive spectroscopy (EDS) analysis on a Nb particle. (**a**) Scanning electron microscopy (SEM) micrograph (SE contrast) with indicated position for EDS line analysis. (**b**) EDS line profile for the elements Nb, O and Al. Element concentrations are shown in at. %.

**Figure 4 materials-12-03927-f004:**
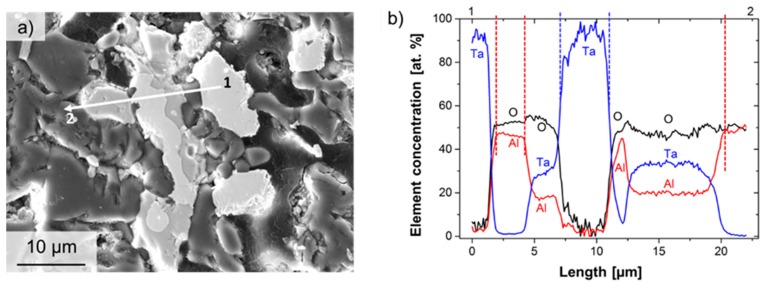
Results of EDS analysis on Ta particles. (**a**) SEM micrograph (SE contrast) with indicated position for EDS line analysis. (**b**) EDS line profile for the elements Ta, O and Al. Element concentrations are shown at. %.

**Figure 5 materials-12-03927-f005:**
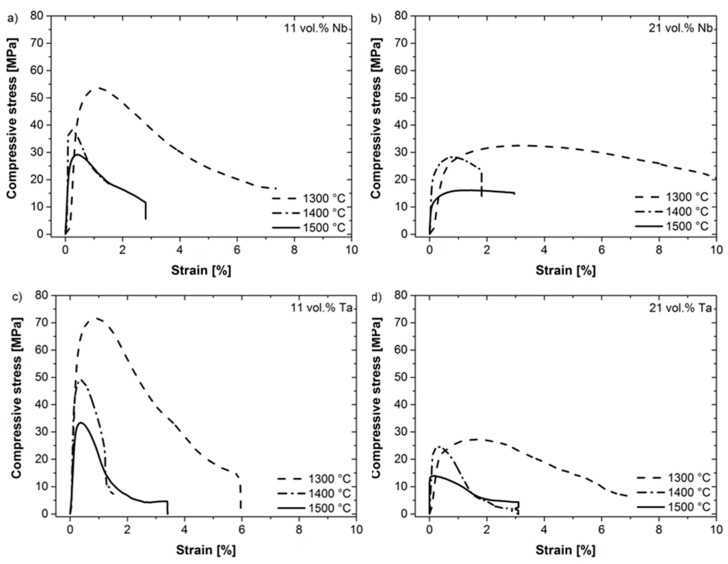
Results of the compression tests up to 1500 °C on coarse-grained Ta-Al_2_O_3_ and Nb-Al_2_O_3_ refractory composites with 11 vol. % (**a,c**) and 21 vol. % (**b,d**) Nb (**a,b**) and Ta (**c,d**), respectively.

**Figure 6 materials-12-03927-f006:**
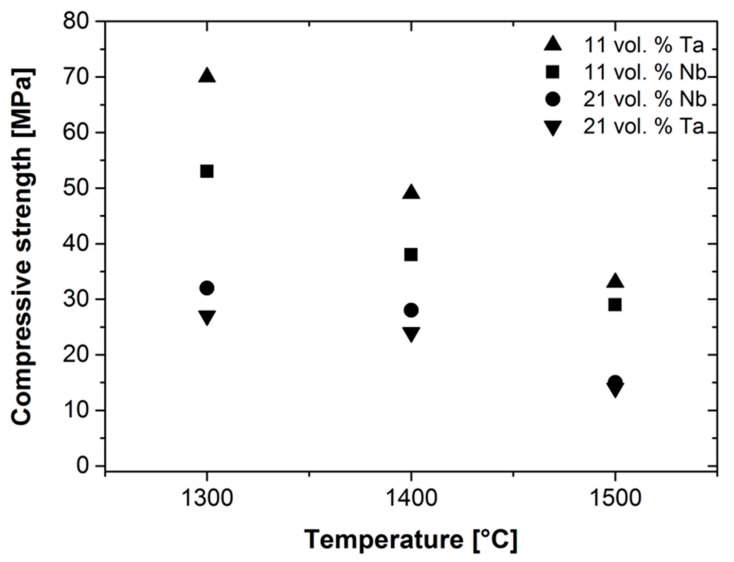
Compressive strengths of the coarse-grained Ta-Al_2_O_3_ and Nb-Al_2_O_3_ specimens at temperatures of 1300, 1400 and 1500 °C.

**Figure 7 materials-12-03927-f007:**
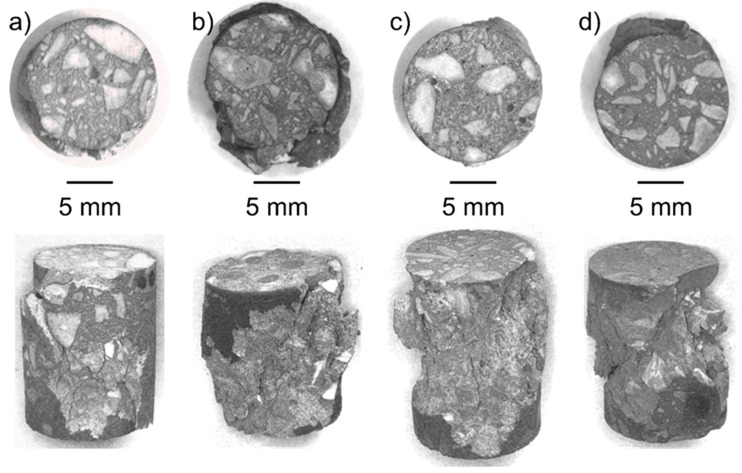
Specimens of coarse-grained Ta-Al_2_O_3_ and Nb-Al_2_O_3_ refractory composite materials with 11 vol. % and 21 vol. % Nb (**a,b**) and Ta (**c,d**), respectively, after compression tests at 1300 °C. Top: Front faces of cylindrical specimens (diameter: 15 mm, height: 22 mm). Bottom: Side views of the cylinders with macroscopic damage pattern.

**Figure 8 materials-12-03927-f008:**
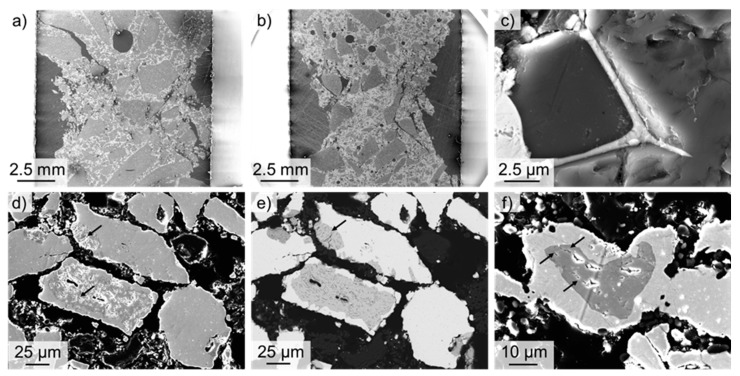
SEM micrographs of specimens after compression tests at 1500 °C. (**a,b**) Overview of the damaged microstructure of specimens with 11 vol. % Nb (**a**) and Ta (**b**). (**c**) Ta enrichment along the grain boundaries of Al_2_O_3_ (21 vol. % Ta). (**d**–**f**) Reaction of Al_2_O_3_ with Nb. (**d,f**) SE contrast. (**e**) BSE contrast.

**Figure 9 materials-12-03927-f009:**
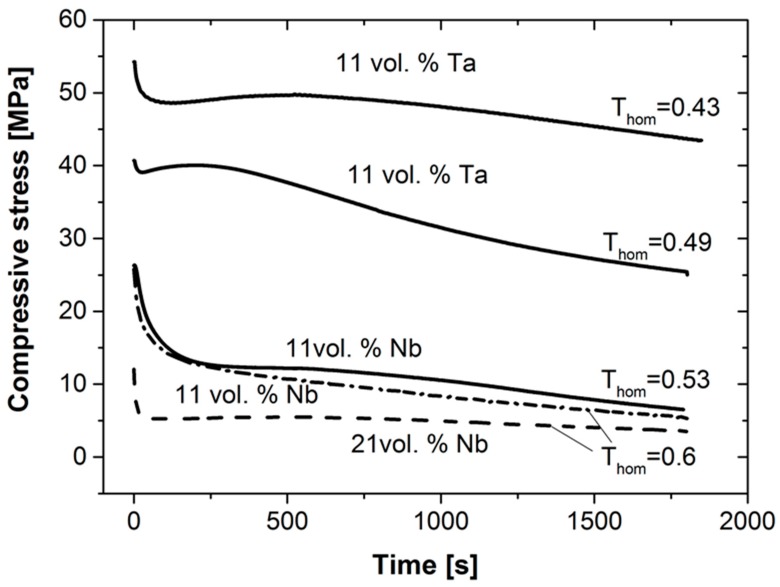
Results of compressive stress-relaxation tests performed on Ta-Al_2_O_3_ and Nb-Al_2_O_3_ specimens at 1300 °C and 1500 °C. The initial compressive loads were chosen with 0.8 of the experimentally determined maximum compressive strengths.

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
