# Peer review of "Mechanical High-Temperature Properties and Damage Behavior of Coarse-Grained Alumina Refractory Metal Composites"

_materials, 2019, doi:10.3390/ma12233927_

Round 1
Reviewer 1 Report
The manuscript is overall well written and presents interesting results. My main concern with this work is that only one mechanical test was carried out for each composition under each condition. It is well known that there is a dispersion in strength values when testing a material, especially in composite materials where the phase distribution and presence of flaws can severely affect the strength values measured. I think that a minimum number of 3 measurements with the corresponding statistical analyses are required to confirm that the trends observed are real.
I also have some minor comments and details that need some attention:
In line 23, the term CMCs is used to refer to Ceramic-Metal composites. This group of materials are usually referred to using the term “cermets”, whereas the term CMC is reserved for Ceramic Matrix Composites. This term has been wrongly used several times throughout the manuscript and could lead to confusion so it should be replaced. In line 25, the main factors limiting the maximum operational temperature of these composites are listed. The thermal mismatch between the ceramic and metal phase is one of these factors and has not been listed, even though it is mentioned later than Ta and Nb have similar CTEs to alumina. Please include this in the text and also provide the values for the CTE of the different phases in the range considered for a better understanding. In line 30 refractory metals with melting points above that of alumina (2054 °C) are listed. The first element listed is Zr which melts at 1855 °C so a correction to that statement should be made to clarify that Zr melts lower than alumina. The paragraph starting in line 61 states that coarse grained ceramic is used opposed to fine grained ceramic due to enhanced creep resistance. After that, there is a mentione of using pre-sintered coarse grains, does this mean that the grains are agglomerates? It is unclear how the ceramic particles are pre-sintered. Please consider re writing this paragraph to clarify. In the materials and methods section, consider using sub-titles instead of including these in the text, for example, line 81 “Materials and Specimen preparation” Please discuss in the text what the criteria were for the selection of the phase content of the composites. There is a mention of 11 and 21 wt.% reinforcement of metals but it seems randomly selected. The castable recipe and the procedure used is described in a previous publication and referenced here. While this is ok, at least describe briefly the procedure to help the reader follow the work. The head displacement rate used during the mechanical testing seems a bit high for a quasi-static test, what was the reason for selected this rate? What is the strain rate experienced by the sample? In the results section, there is a discussion about compound formation at the interface between Ta/Nb and Al2O3. The binary phase diagrams of both metals and O and Al show some solid solubility before any intermetallics or oxides are formed. Have you checked the Ta/Nb/-O-Al ternary diagrams to confirm that the mentioned oxides and tantalates are stable at those temperatures? Does grain growth of the metal phase occur after testing at high temperature? If so, consider including a discussion on how this affects the mechanical response of the composite. From Figure 5 there seem to be slight differences in the materials stiffness with temperature.Author Response
Please see the attchement.

Reviewer 2 Report
Interesting manuscript about the mechanical properties of at some extent new generation of coarse-grained refractory composites based on pre-synthesized coarse grains of a ceramic (Al2O3) and refractory metals (Nb, Ta). The conclusions are well supported by experimental data.
The source, origin, grade of Ta and Nb should be added to text.
Author Response
Comment 1: The source, origin, grade of Ta and Nb should be added to text.
Answer: Done.
Reviewer 3 Report
Dear Authors,
I have read your manuscript carefully and I would say that this manuscript, unfortunately, would be interesting for a very narrow group of readers. The objectives of the study are not sufficiently defined. The introduction provides a good, generalized background of the topic. The results are explained, but not discussed in detail with literature data. Fortunately, they are presented in an appropriate format. The figures show essential data; some of the data are also summarized in the text. I do not think any additional graphics are necessary. The cited literature is relevant to the study and balanced. This manuscript was quite well prepared and could be published after correcting some errors and adding some essential missing information listed below.
- What was the reason for using coarse-grained Al2O3 powder instead of fine powder? In the discussion there is a lack of comparison between the properties of the composites with coarse-grained Al2O3 powders obtained by the authors and literature data of composites with fine powders. This comparison with literature data is needed to indicate not only the novelty of the work, but what is much more important from the practical point of view, the possible areas of application of this type of composites. For this reason, the use of coarse-grained Al2O3 powder does not have a good background.
- What was the reason for selecting the volume contents (11 and 21%) of the Ta and Nb powders? On what basis was the selection made?
- The origin of the powder with the manufacturer/supplier, purity and chemical composition, if they are available, need to be provided.
- In line 93 there is a lack of a diameter symbol. Currently, a different, wrong symbol is used.
- In line 12 in the abstract it would be reasonable to write “The study presents the mechanical properties…” or “The present study shows the mechanical properties…” to avoid double use the word “present”.
Author Response
Please see the attachement.

Reviewer 4 Report
In this work, the mechanical properties of a new coarse-grained refractory composites based on pre-synthesized coarse grains of a ceramic (Al2O3) and refractory metals (Nb, Ta) have been investigated. Before further consideration the following issues should be considered and addressed:
The Abstract is rather short and is not informative. This section should be rewritten including some key findings of the work. Some references for the first paragraph of the introduction should be cited. The words that also their abbreviations have been mentioned, should be capital. i.e. Scanning Electron Microscopy (SEM), etc. In the materials and methods section there are some sub-sections that should be separated from the following paragraph and also be bold. i.e. Materials and specimen preparation. Mechanical testing, etc. Niobium and Tantalum in the entire text should be capital. The same comment of 4 for the results sections. The title of results should be changed to the results and discussion The majority of the references are very old and it is highly recommended to update them.Author Response
Please see the attachement.

Round 2
Reviewer 1 Report
The authors have addressed all the minor comments and suggestions and I believe the quality of the manuscript has improved significantly. However, the main major comment has not been corrected and is still a serious concern.
I mentioned that one of my biggest concerns with this paper is that the authors draw conclusions on mechanical behavior and failure stresses based on a single experiment per experimental condition. The authors mention that the cost of the experiments is a limiting factor and argue that the dispersion of the strength data decreases with the ductility of the material. However their data clearly shows a brittle-dominated fracture (admittedly not catastrophic brittle failure, but not completely ductile either) which is extremely sensitive to the phase distribution and presence of defects within the material. Given the uncontrolled and unforeseen reactions that their material undergoes, I believe that the strength of the material will exhibit a significant dispersion. Furthermore, the change in fracture strength vs. temperature spans over a 50 MPa range which is of the order of magnitude of strength data dispersion reported for similar cermet systems like ZrC/W where the W content was 35 vol% (See Caccia et. Al, Nature 562, 406–409 (2018).
The authors also mention that brittle materials follow Weibull statistics and that an accurate determination of the Weibull modulus requires a minimum of 30 tests. I wouldn't recommend measuring the Weibull modulus because it would be outside of the scope of this work. However, I do believe that some repeats are necessary to validate the conclusions drawn from this study. I think a minimum of 3 repeats per experimental condition is needed to be able to obtain a mean value and an associated standard deviation. If doing repeat experiments is not an option perhaps a contrasting experiment that would confirm the behavior observed in the compression tests could be performed (such as high temperature indentation for example).
Author Response
We submitted our statement according to your serious concern on the validation of the experimental results to the assistant editor. Please, find here a short cutout of our response letter to the assistant editor Joan Zhuang.
We understand the criticism well. However, the manuscript presents brand-new results on a new class of refractory metal-alumina composites. These are first steps in direction of developing a new class of high-temperature refractory materials. Beside the excellent shrinkage behavior, the new materials exhibit also promising mechanical properties as shown in the manuscript. We are sure that the results will obtain broad interest in near future. However, we carried out the study in advance of the establishment of a new larger research project dealing with this topic. Therefore, no financial funding and no project is present for this research currently.
A repetition of mechanical tests for each material and each temperature is excluded and high temperature indentation tests suggested by you are not available in our labs. Moreover, due to the inhomogeneous microstructure of the composite material, we assume that indentation tests are not useful at all.
With kind regards on behalf of all co-authors
Anja Weidner
Reviewer 3 Report
The revised manuscript is suitable for publication in the Materials journal.
Author Response
thank you for your final decision.
Kind regards on behalf of all co-authors.
Anja Weidner